# Sorption Properties of PET Copolyesters and New Approach for Foaming with Filament Extrusion Additive Manufacturing

**DOI:** 10.3390/polym15051138

**Published:** 2023-02-24

**Authors:** Nadiya Sova, Bohdan Savchenko, Victor Beloshenko, Aleksander Slieptsov, Iurii Vozniak

**Affiliations:** 1Department of Applied Ecology, Technology of Polymers and Chemical Fibers, Kyiv National University of Technologies and Design, Nemirovicha Danchenko Street, 2, 01011 Kyiv, Ukraine; 2Donetsk Institute for Physics and Engineering named after O.O. Galkin, National Academy of Sciences of Ukraine, pr. Nauki, 46, 03028 Kyiv, Ukraine; 3Centre of Molecular and Macromolecular Studies, Polish Academy of Sciences, Sienkiewicza Street, 112, 90-363 Lodz, Poland

**Keywords:** polyethylene terephthalate copolyesters, sorption properties, mechanical properties, foaming, additive manufacturing

## Abstract

The mass transfer process of binary esters of acetic acid in polyethylene terephthalate (PET), polyethylene terephthalate with a high degree of glycol modification (PETG), and glycol-modified polycyclohexanedimethylene terephthalate (PCTG) was studied. It was found that the desorption rate of the complex ether at the equilibrium point is significantly lower than the sorption rate. The difference between these rates depends on the type of polyester and temperature and allows the accumulation of ester in the volume of the polyester. For example, the stable content of acetic ester in PETG at 20 °C is 5 wt.%. The remaining ester, which has the properties of a physical blowing agent, was used in the filament extrusion additive manufacturing (AM) process. By varying the technological parameters of the AM process, foams of PETG with densities ranging from 150 to 1000 g/cm^3^ were produced. Unlike conventional polyester foams, the resulting foams are not brittle.

## 1. Introduction

Low-molecular-weight organics are commonly used as solvents in a variety of industrial applications. Their ability to dissolve organic material is very valuable, but solubility is not always desirable, especially when material resistance or permeation properties are involved [1]. The interaction of polymeric materials with solvents may have some important applications [2]. Solubilization of low-molecular-weight polymers is a common basis for coatings [3]. For high-molecular-weight polymers, interaction with solvents is usually a problem of chemical resistance and stability or permeation under the influence of certain media [4].

For high-polymer solvents, the extent of compatibility may vary. Depending on the chemical and molecular structure, solvent diffusion may or may not be limited. When diffusion is not limited, the polymer and solvent form a solution; in contrast, limited diffusion can produce a wide range of structures, such as gel-like structures and plasticized systems. Polymeric materials consist of long molecules organized in some form under the molecular structure, which is a result of the manufacturing process of the thermochemical history of the material. Low-molecular-weight solvents can penetrate the molecular structure by diffusion processes and affect the sub-molecular structure of the polymer [5]. The influence of the solvent on the polymer can be summarized in terms of solubility parameters, thermodynamic compatibility, and intermolecular bonds that refer to specific chemical groups. In general, diffusion without influence on material properties can be described as permeability properties [6], diffusion with influence on intermolecular distance—as plasticization, diffusion with influence on the sub-molecular structure—as solvent crystallization and solvent cracking. The sorption of low-molecular-weight molecules affects the dimensional stability of the material [7].

Polyesters are a widely used and industrially important class of high-molecular-weight polymers. Members of the polyester family include polyethylene terephthalate (PET), polybutylene terephthalate, and polylactide, which are the most important polymers nowadays [8]. From a historical perspective, PET was one of the first industrial polymers and is now one of the most important standard polymers [9]. During the decades of industrial use of PET, polyester has been modified and improved for various end uses [10]. Containers made of PET polyester represent a revolution in the field of liquid foodstuffs, as they have exceptional properties and are easy to manufacture [11]. Moreover, the importance of packaging applications has changed the manufacturing technology of polyester from homopolymers to copolymers, which is now the common name for PET polyester. Further development of PET polyester is aimed at further influencing the crystallization behavior through more complex chemical modifications, leading to the creation of polymers of the PET copolymer family. PET polyester with a high degree of glycol modification, or PETG, is the second member of the family to exist as a stand-alone polymer [12]. PETG polymers with low crystallization rates are the ideal choice for thick-walled transparent packaging. The low crystallization rate is the result of the inhomogeneity of the molecular structure, which affects the folding of long molecules into thermodynamically stable structures. Low crystallization ability leads to amorphous structures with low glass transmission temperature, which is responsible for the thermal stability of the material during the application [13]. Another important member of the PET polyester family is glycol-modified polycyclohexanedimethylene terephthalate (PCTG) [14].

The interaction with volatile organic compounds (VOC) can be a key factor in the final application of the article made from a particular polymer material [15]. Contact sorption and vapor-phase sorption are common types of VOC interaction in real industrial applications [16]. Mass transfer analysis is the most reliable and rapid method to investigate possible interactions between polymer and VOC [17].

Additive manufacturing (AM) is a modern manufacturing approach and an important element of the fourth industrial revolution that is changing traditional manufacturing principles [18]. Adding or patterning materials layer by layer in a programmable manner is an evolutionary approach in many manufacturing industries. Polymeric materials were one of the first applications for additive manufacturing technology, which is now spreading to all industries [19,20,21]. Additive manufacturing is a modern application of PETG that takes advantage of certain material properties. The low melting point and amorphous structure create ideal conditions for cohesion between layers and low shrinkage of parts. Polymer AM uses different materials in different physical forms—liquids, filaments, powders, and granulates [22]. The most widely used process is material extrusion, specifically, melt extrusion and its most widely used type—fused filament fabrication. The main advantages of fused filament fabrication are its wide availability and the simplicity of equipment [23].

AM consists of a physical and programmable technology layer. A unique future of AM is programmable shape transformation into a core-shell structure, which enables the creation of new structures and materials with new property combinations [24]. Programmable transformation offers myriad possibilities for material distribution in the space of the final product and provides a background for new materials and technologies such as metamaterials, shape-memory materials, and smart materials. New AM technologies are emerging and gaining the attention of researchers, such as 4D printing, multi-material printing, controllable shape transformation materials, and many others [25,26,27,28,29]. At the current stage of development of AM technology, there are already so many ways to influence material properties that it is difficult to imagine how to investigate and evaluate all of them. One of the most important possibilities for a programmable part in an AM process is the infill structure or programmable distribution of the material in the volume. The volume of the part can be completely or partially filled with material, allowing control of the direction and distribution of the material flow and the creation of a cellular structure similar to conventional foam.

Foaming is a technology for the production and application of polymer composites with the goal of density reduction, insulation, etc. Generally, foaming technologies are classified according to the method of pore formation. There are two main types of foaming agents to form bubbles: physical and chemical. The most common is a chemical foaming agent, in which the blowing gas is formed as a result of the reaction, e.g., heat-induced decomposition of the chemical substance at the process temperature. Physical foaming agents, classified as either atmospheric gases (such as argon and helium) or as volatile liquids (e.g., propanes or heptanes), are metered directly under pressure into the polymer melt to generate bubbles [30].

In additive manufacturing, foaming technology can be applied as follows: (1) in a programmable way as a filling pattern, and (2) by using materials that can foam during the AM process. Foaming technology for extrusion AM is available as commercial filamentary material with a premixed chemical blowing agent [31,32,33]. The decomposition temperature of the blowing agent is in the same temperature range as material extrusion.

For PET polyester, industrial foaming technology is available as physical foaming with carbon dioxide or fluoro-organic compounds injected under pressure into the polymer melt. The application of supercritical carbon dioxide to PET polyester with extended sorption is used to foam materials [34]. Currently, physical foaming has not been developed for the AM process. In the current study, a special type of foaming is proposed for a series of PET polyester polymers. The application of polymer material with equilibrium VOC can also be used for other polymers. Polymer additive manufacturing uses interaction with solvents and VOC to smooth and bond the surface of parts [35,36,37,38,39,40].

The aim of the present study is to investigate the influence of binary acetic acid esters on the long- and short-term properties of common and modern PET polyesters. The investigation of the long-term mass transfer properties of polyester materials leads to the identification of a specific material property and its perspective application in a common additive manufacturing process.

## 2. Materials and Methods

### 2.1. Materials

Acetic esters—ethyl acetate and butyl acetate—were supplied by Telco LLC, Kyiv, Ukraine as commercial purity solvents. Glycol-modified PET polyester Skygreen^®^ KN100—PETG, PCTG Skygreen^®^ JN100, and PET copolymer Skypet^®^ BR were supplied by Biesterfeld Special Chem, Kyiv, Ukraine.

### 2.2. Experimental Setup

Sorption of acetic acid esters in the polyester polymers was carried out in saturated vapor and in direct liquid contact under stabilized temperature conditions in a temperature-controlled chamber. Vapor sorption was carried out in a closed glass exicator in a saturated vapor medium directly under the liquid phase.

The polyester polymers were converted into films and monofilaments using laboratory extrusion equipment. Prior to extrusion, the material was dried in a Memmert ULP 500 convection oven.

PET copolymer—drying time 10 h, drying temperature −160 °C. PETG and PCTG polyesters—drying time 8 h, drying temperature 60 °C.

The tape samples were extruded through a flat slit die into a hot water bath in the vertical direction. The samples were cooled in the hot water bath to achieve the fastest possible cooling. The water bath temperature was 70 °C. The extrusion rate of the material was 1.8 kg/h. Before further experiments, the tapes were measured with a contact micrometer to check the uniformity of the thickness with reasonable tolerances in the range of ±1.5%. The spinneret draw ratio of 200% was kept constant for all extruded materials. The temperature profile of the extrusion line was constant for all materials and was 240–275–260–265 °C from the first heating zone to the die head. A single screw extruder with a diameter of 27 mm and a ratio of 30 L/D was used with a speed of 22 to 26 rpm depending on the bulk density of the material and constant mass productivity.

Monofilament extrusion was carried out in the horizontal direction with the same extruder setup and a round spinneret head. The melt was extruded through a 2.5 mm round die with an L/D ratio of 10. The draw ratio of the spinneret was kept in the range of 200 to 250% to obtain the desired filament diameter. The melt was cooled in a horizontal water bath with forced hot water circulation. Water bath temperature 70 °C. The extrusion rate of the material was 1.2 kg/h. The extruded samples were wound onto a plastic spool with as little tension as possible.

The film specimens were used for punching out standard specimens for tensile testing. Samples and cutting blade were preheated to 60 °C before the cutting procedure in order to minimize edge cracking.

### 2.3. Characterization

The samples of polyester material were prepared by melt extrusion and cooling. The sample was cooled at a high cooling rate to obtain an amorphous structure. The material density was measured by hydrostatic weighing.

Sorption and desorption of acetic acid esters were studied by weighing the samples over a time scale. Samples for sorption were cut to a size of 20 × 20 mm with a thickness of 0.5–0.6 mm.

Liquid-phase sorption was performed by immersing the polyester sample in a glass containing ester, which was placed in a temperature-controlled chamber. The jar containing the ester and the sample was sealed with a lid to prevent evaporation of the ester and heat absorption. To observe the sorption kinetics, the samples were accurately weighed after exclusion from the liquid phase.

The desorption was carried out under ambient conditions, elevated temperature, and reduced pressure in the vacuum-drying chamber LMM LP40412. The desorption of acetic ester was performed under a normal atmosphere in a temperature-controlled chamber. Forced desorption was carried out in a forced-air oven at an elevated temperature and in the vacuum-drying chamber at reduced pressure. Polyester samples were studied at different stages of sorption and desorption. Mechanical properties—tensile strength and elongation at the break—were measured on standardized specimens according to ISO 527-2:2012 [41]. Shore D hardness was measured on the same specimens with a thickness of 4 mm ISO 7619-1:2010 [42]. Material density was measured by hydrostatic weighing in water ISO 1183-1:2019 [43]. The relative error in the determination of density was not more than 2%. The melt flow rate was measured according to ISO 1133 [44] using a capillary of 2.095 mm diameter and weight of 2.16 kg.

The material samples were analyzed by surface FTIR using Perkin Elmer Spectrum 3 infrared spectrometer according to ASTM D5477—18 [45]. Thermal analysis was conducted using DSC TA devices, such as DSC 2920 according to ISO 11357-1:2009 [46]; the heating rate was 5 °C per minute. The material structure was evaluated by SEM JEOL JSM −5500 LV.

## 3. Results and Discussions

### 3.1. Liquid-Phase and Vapor-Phase Sorption of Ethyl Acetate for Polyesters

Acetic ester is a common ingredient in industrial solvents and some detergent formulations. When developing product packaging, the chemical formulation of the product must be tested for short- and long-term compatibility with the polymer material of the packaging. In routine compatibility testing of PETG polyester, the authors observed that limited sorption of ethyl acetate occurs in liquid phase contact sorption after a prolonged period of time. This limited sorption leads to swelling of the material and changes in hardness and flexibility. Acetic acid esters of different molecular weight—ethyl acetate and butyl acetate—were used as sorption media in the liquid and vapor phases.

Liquid-phase sorption of ethyl acetate is concentration-limited and has a stable maximum extent that is temperature dependent. At 20 °C sorption extent reaches 19.2% for PETG, 18.04 wt.% for PCTG, and 16.0 wt.% for PET. Temperature increase from 20 °C to 40 °C results in increasing the sorption extent to 25%. The PETG exhibits the highest value of consumed ester for both temperature set points, probably due to the higher content of amorphous phase among the polyesters studied.

An important observation in liquid-phase sorption is an intense cohesive bonding of the material surfaces, which is probably due to the self-healing effect of the surface and the plasticization effect. This effect is known for most semi-crystalline polyesters when reaching or exceeding the glass transmission temperature [47]. The main reason for cohesive bonding is molecular mobility at the surface, which may be favored by temperature or sorption of low-molecular-weight substances.

Another observed effect is a change in sample color and light transmission, with transparent samples changing their base color to opalescent white due to possible propagation of the solvent-induced crystallization process (Figure 1).

The sorption of ethyl acetate in the vapor phase at 20 °C is shown in Figure 2a. The sorption maximum in the vapor phase is lower by 12–14% than in the liquid phase for all the polyesters studied. An increase in temperature from 20 °C to 40 °C leads to an increase in the maximum extent sorption for PETG by 14%, for PET by 1%, and for PCTG by 3%.

To describe the influence of the molecular weight of the acetic acid ester, the acetic acid ester with a higher molecular weight such as butyl acetate was used for comparison (Figure 2b). Butyl acetate exhibits different kinetics and extent. The sorption of butyl acetate has a two-step equilibrium behavior, the sorption rate is lower compared with ethyl acetate, but the maximum sorption extent at equilibrium is much higher. Butyl acetate has a higher molecular weight than ethyl acetate; however, both have a high maximum sorption degree. This observation may be related to molecular conformation and compatibility.

Due to the condensation production process, polyesters are known for their affinity for water vapor sorption [48]. The influence of initial water content in polyester was studied in the vapor sorption of ethyl acetate and polyester samples in the ambient and dried conditioned state. The water content was determined by weighing the samples before and after the vacuum-drying process at 40 °C and 10 mbar pressure, and it was 0% and 0.32%.

Dried samples show higher maximum sorption and sorption rate: for PETG by 15%, for PET by 1.6%, and for PCTG by 3.5% higher than for standard samples. These results can be explained by the higher affinity of the solid molecular structure of the polyester for the smaller water molecules and the same type of polar intermolecular interaction. The increase in the maximum sorption range for dried samples is significantly higher than the water content in conditioned samples. Thus, water has some simultaneous sorption with polyesters and influences the extent of sorption of acetic acid esters.

The rate and extent of sorption are the highest in the first 24 h of the sorption process (Figure 3). There is little change with respect to sample weight during the first two hours, and the rapid mass transfer begins after this induction period. This behavior can possibly be explained by the surface penetration of acetic ester.

### 3.2. Desorption of Acetic Esters from Polyesters

Limited sorption of acetic esters in the saturated vapor is reversed by evaporation when the sample is stored at ambient conditions. Desorption of acetic esters was carried out under ambient conditions and at elevated temperatures in a temperature-controlled chamber with natural air circulation.

It was found that the desorption of acetic esters from polyester under ambient conditions and at elevated temperature is partial or incomplete, which is called incomplete desorption (Figure 4a). The desorption rate is significantly lower than the sorption rate for all polymers and esters studied. Butyl acetate shows a twofold desorption curve (Figure 4b). The desorption does not reach its full value for all the polyesters studied and shows a stable equilibrium value over time. The extent of residual equilibrium desorption is temperature-dependent for all polyesters and esters studied.

A long-term study of the residual ester content shows a very slow decrease in ester content over time. At 20 °C and ambient pressure, the ester content can decrease to 0.9% by weight after one year of desorption. The slow rate of desorption in the equilibrium zone is significantly affected by temperature. At 60 °C, complete desorption for PETG and ethyl acetate can be achieved after 27 days. At 60 °C and vacuum (pressure 10 kPa), complete desorption for the same system is achieved in 36 h. For other polyesters, complete desorption can be reached in a similar way but within a different time period. The desorption of esters from polyesters is relatively slow, which offers the possibility to control the residual ester content by choosing specific conditions for the desorption process.

### 3.3. Influence of Sorption of Acetic Acid Esters on Material Properties

The process of sorption of selected acetic acid esters is accompanied by dramatic changes in the material physical properties (Figure 5 and Figure 6). Sorption is found to be accompanied by swelling of the polyesters and a decrease in their hardness, which is due to the plasticizing effect of the esters on the polyesters. The behavior of density correlates with the behavior of hardness.

The tensile properties of polyesters at different stages of sorption and desorption of acetic ester were studied on tape and filament samples. The following measurements were carried out on the samples: under steady-state conditions—initial condition, conditions of maximum sorption extent; and under stable conditions—incomplete desorption and conditions of complete desorption. The results are listed in Table 1.

The mechanical properties at the maximum sorption level are characterized by a significant increase in tensile elongation and a decrease in tensile strength and tensile modulus, which can be attributed to the plasticization effect. Samples with incomplete desorption of the ester show similar behavior, indicating a moderate decrease in mechanical properties. Compared with the original properties, the level of properties of materials with incomplete desorption is suitable for some functional applications.

Materials after complete desorption of the ether have a low tensile elongation and a higher tensile modulus than the original material. The change in the current properties can be explained by the crystallization caused by the ether. All polyesters used in the current study are semi-crystalline polymers, so sorption of the ester and swelling may cause an increase in molecular mobility, which may facilitate the crystallization process at elevated temperatures.

FTIR spectroscopy and DSC studies were performed for PETG. The FTIR spectra of PETG polyester under stable sorption conditions are shown in Appendix A. Compared with the initial sample, sorption of the ester resulted in a decrease in absorbance at the 3400 cm^−1^ band, which can be attributed to the terminal hydroxyl group of the polyester. One possible explanation for this behavior is hydrogen bonding between the ester and polyester. No other discernible changes in the spectrum were observed.

To investigate the effects of ester mass transfer on the amorphous and crystalline phases of the materials, thermal analyses by DSC were performed on the initial samples and on the samples after the complete desorption of the ester (Figure 7).

The DSC heating curves of the original PETG and PETG after the complete desorption of the ester show an increase in the glass transition temperature from 73 to 78 °C. The increase in glass transition temperature and the appearance of a more pronounced melting peak can be explained by the contribution of ester mass transfer through the material to the formation of a crystalline structure in PETG.

### 3.4. Melt Processing and Additive Manufacturing

The melt index was measured on samples of polyesters in the initial state and the state of complete desorption after appropriate drying. The results shown in Table 2 indicate that there is essentially no change in melt viscosity after the desorption of the ester from the polyester.

Melt index measurement experiments performed on polyester in the incomplete desorption state show that the material can physically foam during melting and extrusion. By filling such material into the heated chamber, fine foam can be extruded from the capillary nozzle of the melt indexer. This experiment was the starting point for the idea of using ester sorption for physical foaming. Filament samples were introduced into a standard 3D printer and passed through an extruder device. The results obtained in this way confirm the general possibility of the proposed idea.

PETG polyester was selected because of the high degree of equilibrium desorption at room temperature. Ethyl acetate was chosen primarily because its odor is relatively acceptable for indoor use, while the odor of butyl acetate is irritating. Filaments of PETG polyester were extruded with a diameter of 1.62 mm. The PETG filament was wound on a metal spool and dried at 60 °C for 8 h in a forced-air chamber. Sorption of acetic ester was performed in a glass exicator in the vapor phase. The sorption temperature was set at 20 °C to reduce the sorption rate. Sorption at room temperature and higher temperatures is associated with intense cohesive bonding of the material. For filamentary material wound on a spool, this leads to complete adhesion of the filament in many places and difficulties in unwinding. Experimental work provides several ways to eliminate these problems—e.g., by reducing the sorption rate and using a surface treatment of the filament. Surface treatment with potassium soap, silicone oil, and talcum powder is used with some results. The use of a very low winding tension along with a low sorption temperature is the simplest solution to the bonding problem. Filaments with an initial diameter of 1.62 mm absorb 5.5% ethyl acetate after 9 h of sorption at 20 °C and swell to 1.77 mm diameter. The process diagram and parameters for filament fabrication are shown in Figure 8 and Appendix A.

Additive manufacturing with the filament PETG (ethyl acetate) was performed on a standard Prusa 3D printer, model I3, without any modification. The model of the article with the specific shape was prepared for the AM process in the slicing software Utilmaker Cura 4.1.

The most successful (from the authors’ point of view) process and slicing parameters determined during numerous experimental trials are listed in Table 3.

It was experimentally found that the physical foaming of PETG is accompanied by high swelling of the extrudate, which can reach 500–800% of the original die diameter. The AM process can be carried out at a relatively high printing speed and high temperature. At a low printing speed, most of the ester can escape from the melt zone of the extruder. Compared with the AM process with normal filament, a very high extrusion height and width are used. Nevertheless, width values are able to maintain adequate interlayer cohesion and article consistency. Foam density can be regulated by printing speed (material residual time), die temperature, and extrusion feed coefficient. Extrusion height and width can also significantly affect material density and structure due to the mechanical action of the heated die on the over-expanded foam.

At ambient conditions and the type of material used, the equilibrium desorption value is 6%. This value is sufficient for the foaming process. Experimental tests have shown that ester content of at least 3% is required for successful foaming during the AM process. Regulation of the ester content can be easily achieved by desorption in a vacuum chamber.

It was found that filaments with ethyl acetate can be successfully melted in the extruder die without significant evaporation of ester. Molten polyester with ethyl acetate is physically formed during extrusion through the orifice of the spinneret. The expansion of the foam at the spinneret is significant and suitable for additive manufacturing of foam articles. It is possible to produce foam articles with low density. The polyester foam stream has strong cohesion between layers and streams. The structure of the foam still contains sorbet ester, even after being produced in the molten state. Compared with other foamed materials, PETG foam is not brittle and can be bent to a high degree without cracking.

The foam specimens were tested for tensile strength in two conditions—as printed and after complete desorption of the ester residues in the vacuum chamber. Complete desorption significantly affects the complex properties of the material (Table 4). The specimens for tensile testing were 3D-printed with two material flow orientations (Figure 9), which is a standard procedure for such specimens in many scientific papers. The specimens were 3D-printed with three different density values (Table 4) by adjusting extrusion height and width, extrusion coefficient, and speed with one material and the same ester content. The material properties are listed for general information and cannot be directly compared due to the different printing settings. By changing the pressure parameters, it is possible to achieve a wide range of material density.

The appearance of the surface of foamed PETG in the optical observation camera is shown in Appendix A. Scanning electron microscope images are shown in Figure 10.

### 3.5. Process Limitations and Drawbacks

The most important process limitation is material cohesion or sticking. For filamentary material wound under tension, this results in strong cohesive bonding and limited possibility of demolding. Another special material property is cracking, which occurs after contact with the ester. The filament material shatters at irregular intervals, especially at high winding tension.

A possible solution to this limitation is the goal of further investigation. A possible solution, already found by the authors, is the influence on material orientation during filament production or post-orientation in the solid state.

## 4. Conclusions

Sorption of esters on polyesters causes plasticizing effect and promotes structural transformation by crystallization. The desorption of ethyl acetate and butyl acetate has a stable equilibrium state, the extent of which depends on temperature, and the rate of desorption is extremely low under these conditions. The sorption of acetic ester affects the complex physical and mechanical properties of materials in a manner typical of plasticization. The sorption equilibrium conditions are stable over a long period of time, which opens the possibility of the practical application of this material property.

PETG polyester with ethyl acetate in the equilibrium state of sorption was used for the extrusion based additive manufacturing process. The remaining ester acts as a physical foaming agent during melt formation and deposition. The conversion of PETG—ethyl acetate in the melt, which is accompanied by foaming, does not result in the complete evacuation of the ester from the polyester. The desorption of the ester from the polyester is relatively slow even in the molten state, allowing the foaming process to occur. The properties of PETG foam are determined by the presence of ester residues. The printed foam samples are soft and not brittle. Foam samples after the complete desorption of the ester show typical brittle foam properties.

## Figures and Tables

**Figure 1 polymers-15-01138-f001:**
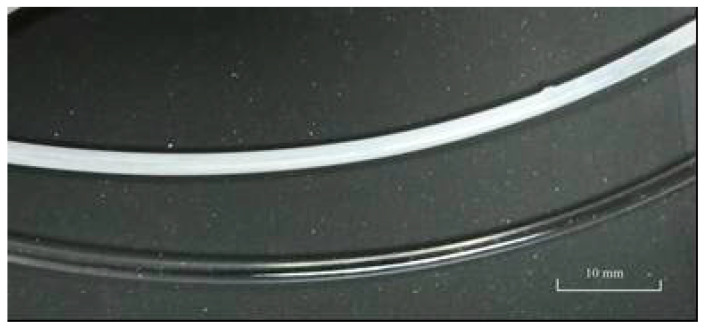
The appearance of PETG polyester sample before and during sorption of acetic ester.

**Figure 2 polymers-15-01138-f002:**
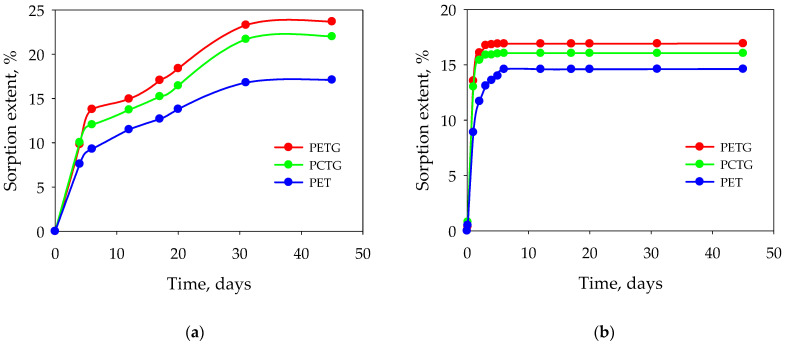
Sorption in the vapor phase: (**a**)—ethyl acetate, (**b**)—butyl acetate.

**Figure 3 polymers-15-01138-f003:**
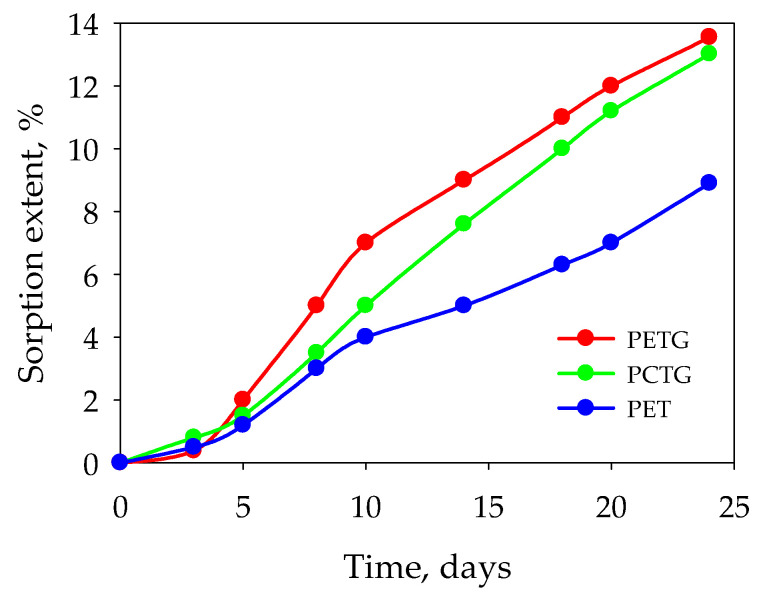
Sorption in the vapor phase of ethyl acetate. T = 20 °C in the first 24 h.

**Figure 4 polymers-15-01138-f004:**
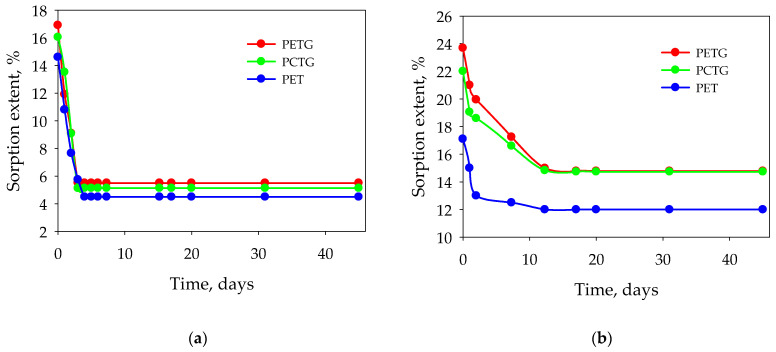
Desorption of acetic acid esters from polyesters: (**a**)—ethyl acetate, (**b**)—butyl acetate.

**Figure 5 polymers-15-01138-f005:**
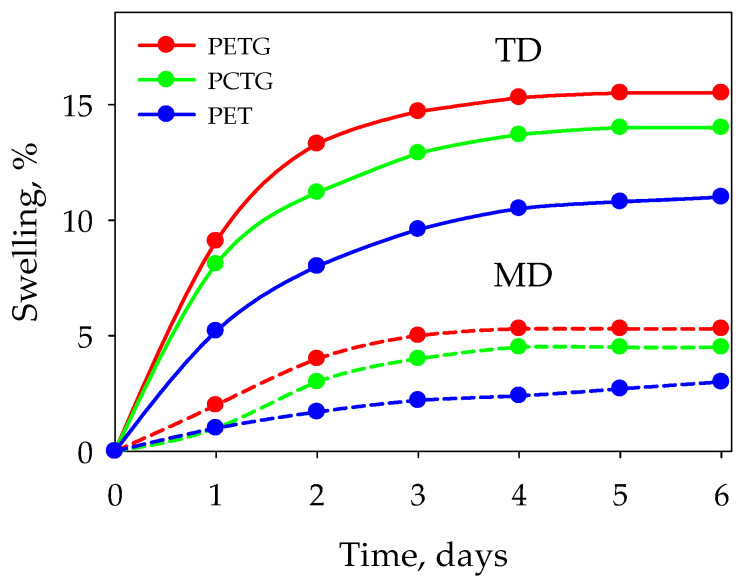
Material swelling during the ester sorption. MD—machine direction, TD—transverse direction.

**Figure 6 polymers-15-01138-f006:**
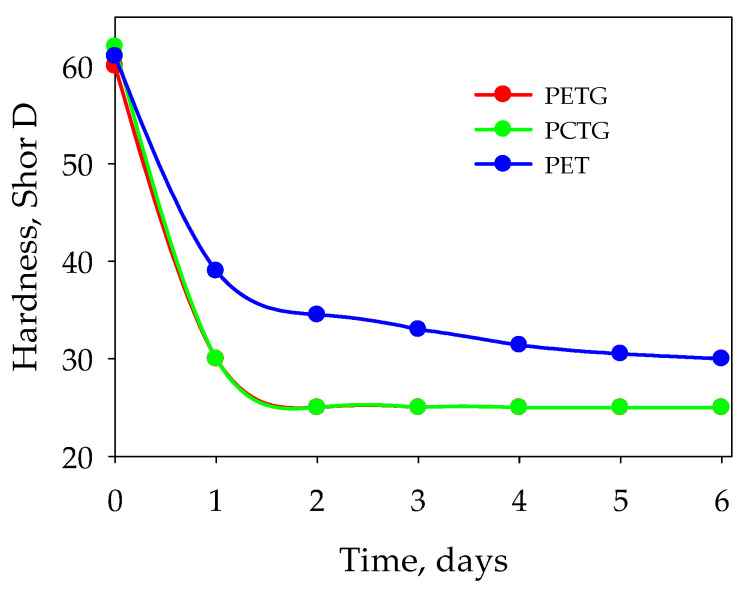
Material hardness during the sorption of ethyl acetate.

**Figure 7 polymers-15-01138-f007:**
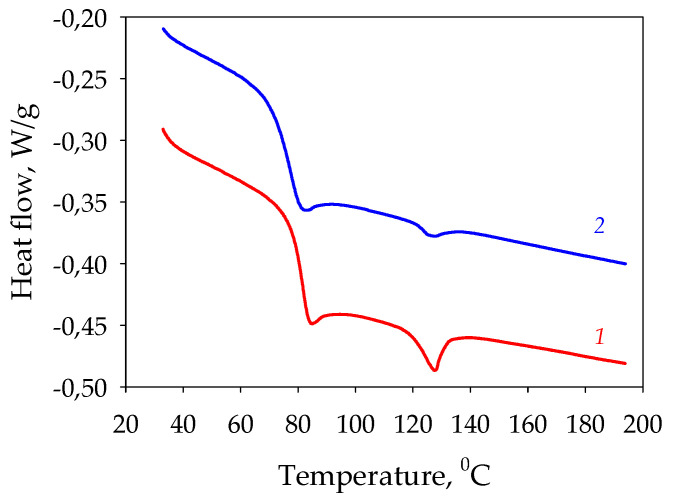
DSC of PETG—ethyl acetate: *1*—sample in equilibrium desorption state, *2*—initial sample.

**Figure 8 polymers-15-01138-f008:**
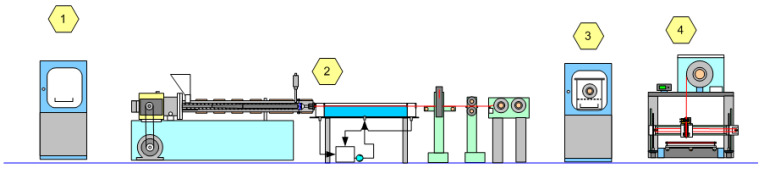
Principal process diagram of AM with PETG foaming: 1—initial material drying, 2—filament extrusion, 3—sorption of acetic acid ester, 4—additive manufacturing.

**Figure 9 polymers-15-01138-f009:**
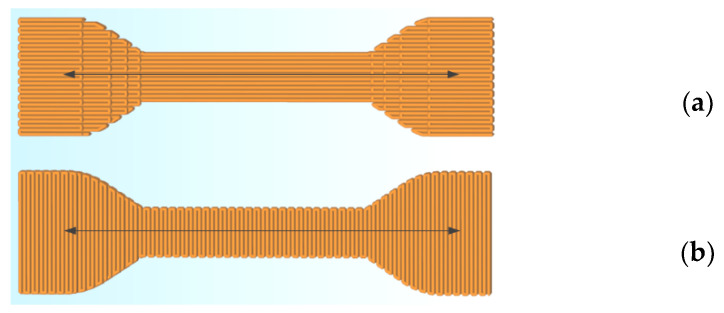
Material stream orientation: (**a**)—Y direction, (**b**) X—direction.

**Figure 10 polymers-15-01138-f010:**
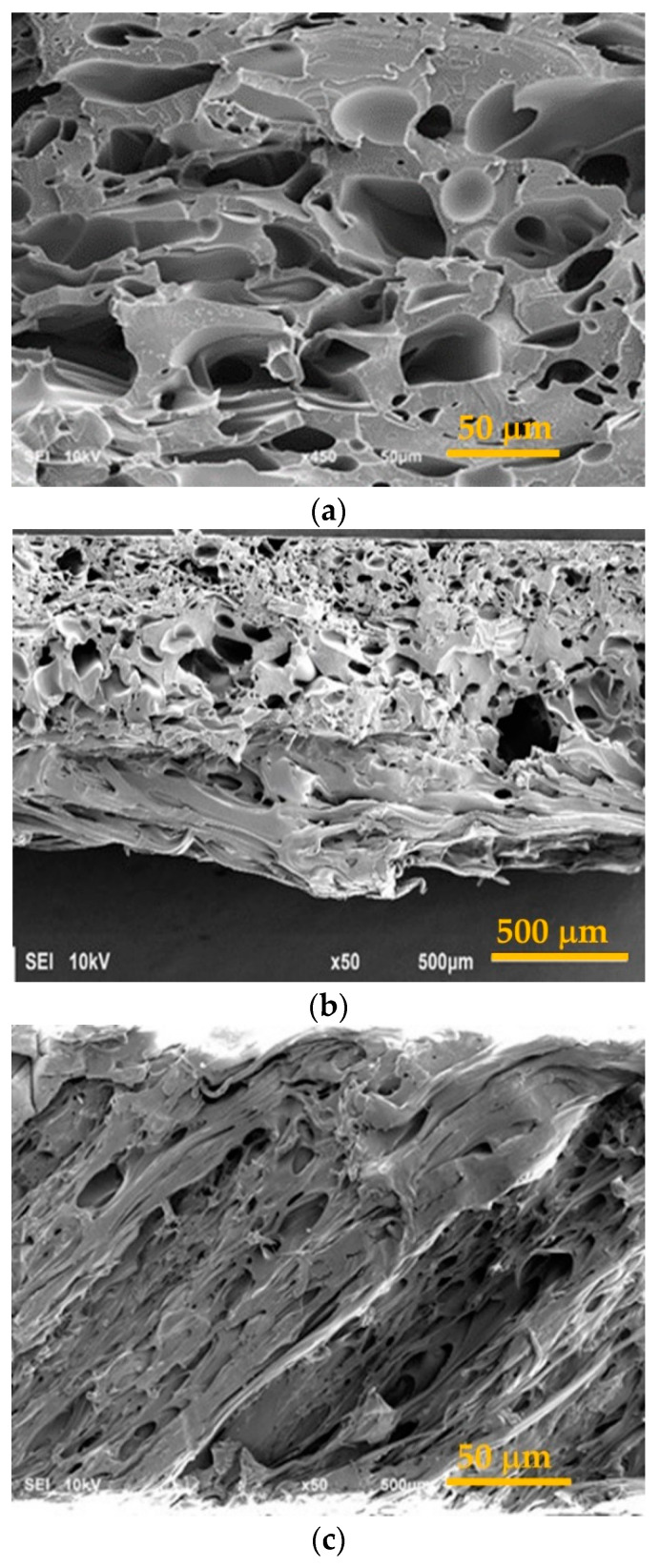
SEM image of PETG foam structure of a sample with different densities: (**a**)—124 kg/m^3^, (**b**)—325 kg/m^3^, (**c**)—650 kg/m^3^.

**Table 1 polymers-15-01138-t001:** Mechanical properties of polyesters under stable sorption conditions (the table shows average values).

Polyesters		Tensile Strength, MPa	Tensile Elongation, %	Tensile Modulus, MPa	Density, kg/m^3^	Hardness, Shore D Scale
Initial	PETG	68 ± 2	32 ± 2	2160 ± 108	1264	60 ± 3
	PCTG	70 ± 2	48 ± 2	2060 ± 103	1245	62 ± 2
	PET	72 ± 2	56 ± 1	2240 ± 112	1340	61 ± 3
Maximum sorption	PETG	38 ± 2	158 ± 8	240 ± 22	1234	25 ± 1
	PCTG	22 ± 1	370 ± 18	150 ± 13	1220	25 ± 2
	PET	34 ± 2	320 ± 16	230 ± 21	1331	30 ± 2
Incomplete desorption	PETG	53 ± 4	54 ± 3	1350 ± 130	1299	54 ± 3
	PCTG	48 ± 3	220 ± 11	1450 ± 142	1224	57 ± 3
	PET	51 ± 4	200 ± 10	1540 ± 150	1380	56 ± 3
Full desorption	PETG	56 ± 4	14 ± 1	2050 ± 200	1269	61 ± 3
	PCTG	62 ± 5	27 ± 2	2110 ± 210	1253	63 ± 3
	PET	61 ± 5	21 ± 1	2360 ± 205	1356	64 ± 3

**Table 2 polymers-15-01138-t002:** Melt flow index of polyester samples at 260 °C, 2.16 kg.

Polyester Type	Initial Sample	After Full Desorption of Ester
PETG	16.4	16.8
PCTG	17.2	17.6
PET	9.5	9.2

**Table 3 polymers-15-01138-t003:** AM and slicing process parameters for PETG foaming.

Process Parameter	Value
Nozzle diameter, mm	0.4
Extrusion width, mm	2.5
Layer height, mm	1.5
Infill rate	100
Extrusion rate, %	80
Nozzle temperature, °C	260
Built platform temperature, °C	60
Printing speed, mm/min	4000
First layer speed, %	50
Air cooling	100%
Retraction, mm	1

**Table 4 polymers-15-01138-t004:** Typical properties of PETG foam.

Property	Value for PETG Ester Modified Foam Sample					
	1		2		3	
	a	b	a	b	a	b
Material density, kg/m^3^	124	118	325	315	650	643
Tensile strength X direction, MPa	4.0 ± 0.3	4.0 ± 0.4	9.0 ± 0.6	8.0 ± 0.4	11.0 ± 0.6	10.0 ± 0.7
Tensile strength Y direction, MPa	5.0 ± 0.3	6.0 ± 0.5	12.0 ± 0.7	13.0 ± 0.7	17.0 ± 1.1	16.0 ± 1.1
Tensile elongation X, %	9.0 ± 0.6	3.0 ± 0.2	12.0 ± 0.7	4.0 ± 0.3	16.0 ± 1.0	4.0 ± 0.3
Tensile elongation Y, %	12.0 ± 0.9	4.0 ± 0.3	16.0 ± 1.2	5.0 ± 0.3	21.0 ± 1.3	4.0 ± 0.3
Ester content, %	1.46	0	1.56	0	1.87	0

a—as a printed sample; b—after full desorption under vacuum.

## Data Availability

Not applicable.

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
