# Peer review of "Sorption Properties of PET Copolyesters and New Approach for Foaming with Filament Extrusion Additive Manufacturing"

_polymers, 2023, doi:10.3390/polym15051138_

Round 1

Reviewer 1 Report

This manuscript studies a new technique for extrusion additive manufacturing foaming processing of copolyester. The subject is interesting and the manuscript has been well written. However, it needs some revisions before final decision.

1. The abstract needs a major revision. Abstracts for research papers should provide the reader with a quick overview of the entire study. Abstracts contain the following elements: i) importance of the topic and/or reference to the current literature and/or identification of a knowledge gap; ii) aim(s) of the current study; iii) indication of the methods used; iv) statement of the key finding(s) and v) implications of the findings and/or value of the current study. However, the Abstract in its current format, contains only the third element.

2. Please avoid using acronyms in the keywords.

3. Please provide the unit of temperature in line 149.

4. Please avoid missing spaces between values and units, for instance, in line 188, line 221, line 223 and etc.

5. Please provide the standard deviations for the results presented in Table 1. And also, for Table 3.

Author Response

Review 1

This manuscript studies a new technique for extrusion additive manufacturing foaming processing of copolyester. The subject is interesting and the manuscript has been well written. However, it needs some revisions before final decision.

  1. The abstract needs a major revision. Abstracts for research papers should provide the reader with a quick overview of the entire study. Abstracts contain the following elements: i) importance of the topic and/or reference to the current literature and/or identification of a knowledge gap; ii) aim(s) of the current study; iii) indication of the methods used; iv) statement of the key finding(s) and v) implications of the findings and/or value of the current study. However, the Abstract in its current format, contains only the third element.

Abstract has been changed.

  1. Please avoid using acronyms in the keywords.

Keywords have been modified.

  1. Please provide the unit of temperature in line 149.

The unit has been added.

  1. Please avoid missing spaces between values and units, for instance, in line 188, line 221, line 223 and etc.

Corrected

  1. Please provide the standard deviations for the results presented in Table 1. And also, for Table 3.

Standard deviations have been added.

Reviewer 2 Report

The title, abstract, and keywords need major revisions.

The title is very general.

The abstract should appeal to the reader. Novelty should be presented transparently. Research achievements should be mentioned quantitatively. The purpose of doing the work should also be mentioned. The current version presents generalities, research methods, and some qualitative results.

Most parts of the introduction are general, especially the third paragraph. The introduction can be written more interestingly and deeper.

For this purpose, it is suggested to use the following new resources in the field of PETG printing.

The introduction is very incomplete and should be presented in more depth. It is suggested to use these resources to improve the 3D printing section. (A comprehensive experimental investigation on 4D printing of PET-G under bending, Assessment of controllable shape transformation, potential applications, and tensile shape memory properties of 3D printed PETG, 4D printing of PET-G via FDM including tailormade excess third shape, Shape memory performance of PETG 4D printed parts under compression in cold, warm, and hot programming).

Has the sample been printed? The most important part of printing new materials with FDM is choosing the optimal printing parameters. While the printing parameters are not mentioned at all. This section needs fundamental reforms. First, the selected parameters are summarized in a table. Then the image of the printed samples will be presented and how to choose the parameters will be explained.

The images in Figure 10 are crude and need to be improved.

Add an error bar to the quantitive results (Tables 1-3)

How are the reproducibility of mechanical properties results checked?

In the results section, in addition to presenting experimental data, discussion and analysis should also be added to it.

The conclusion should be modified as in the abstract section.

Author Response

Review 2

The title, abstract, and keywords need major revisions.

Abstract, title and keywords have been changed.

The title is very general.

The new title has been proposed - Sorption properties of PET copolyesters and new approach for foaming with filament extrusion additive manufacturing.

The abstract should appeal to the reader. Novelty should be presented transparently. Research achievements should be mentioned quantitatively. The purpose of doing the work should also be mentioned. The current version presents generalities, research methods, and some qualitative results.

Abstract has been rewritten.

Most parts of the introduction are general, especially the third paragraph. The introduction can be written more interestingly and deeper.

For this purpose, it is suggested to use the following new resources in the field of PETG printing.

The introduction is very incomplete and should be presented in more depth. It is suggested to use these resources to improve the 3D printing section. (A comprehensive experimental investigation on 4D printing of PET-G under bending, Assessment of controllable shape transformation, potential applications, and tensile shape memory properties of 3D printed PETG, 4D printing of PET-G via FDM including tailormade excess third shape, Shape memory performance of PETG 4D printed parts under compression in cold, warm, and hot programming).

The introduction has been changed and updated.

Has the sample been printed?

Samples were printed and tested. A picture of the printed parts is in Fig. S2 in the "supplementary materials" section.

The most important part of printing new materials with FDM is choosing the optimal printing parameters. While the printing parameters are not mentioned at all. This section needs fundamental reforms. First, the selected parameters are summarized in a table. Then the image of the printed samples will be presented and how to choose the parameters will be explained.

The table of AM and slicing process parameters for PETG foaming was in the Supplementary Materials section in the previous version of the manuscript; in the revised version of the manuscript, the table has been moved to the main text (Table 3).

The images in Figure 10 are crude and need to be improved.

SEM Images have been improved.

Add an error bar to the quantitive results (Tables 1-3)

Standard deviations have been added.

How are the reproducibility of mechanical properties results checked?

Reproducibility was ensured by using standard statistical analysis and appropriate sampling.

In the results section, in addition to presenting experimental data, discussion and analysis should also be added to it.

Results section has been expanded.

The conclusion should be modified as in the abstract section.

In the revised version of the manuscript, the conclusions correspond to the abstract.

Round 2

Reviewer 2 Report

Almost most of the comments have been well-answered and considered. Only in the introduction section, it is better to use more sources that focus on PETG. Refer to the comments of the previous step.

Author Response

Comments and Suggestions for Authors

Almost most of the comments have been well-answered and considered. Only in the introduction section, it is better to use more sources that focus on PETG. Refer to the comments of the previous step.

We have slightly revised the introduction section and added the following references to the papers related to the 4D printing of PET.

Aberoumand, M.; Soltanmohammadi, K.; Soleyman, E.; Rahmatabadi, D.; Ghasemi, I.; Baniassadi, M.; Abrinia, K.; Baghani, M. A comprehensive experimental investigation on 4D printing of PET-G under bending. J. Mater. Res. Technol., 2022, 18, 2552-2569.

Soleyman, E.; Aberoumand, M.; Rahmatabadi, D.; Soltanmohammadi, K. ; Ghasemi, I.; Baniassadi, M.; Abrinia, K.; Baghani, M. Assessment of controllable shape transformation, potential applications, and tensile shape memory properties of 3D printed PETG. J. Mater. Res. Technol., 2022, 18, 4201-4215.

Soleyman, E.; Rahmatabadi, D.; Soltanmohammadi, K.; Aberoumand, M.; Ghasemi, I.; Abrinia, K.; Baniassadi, M.; Wang, K.; Baghani, M. Shape memory performance of PETG 4D printed parts under compression in cold, warm, and hot programming. Smart Mater Struct, 2022, 31(8), 085002.

Niazy, D.; Elsabbagh, A.; Ismail, M.R. Mono–Material 4D Printing of Digital Shape–Memory Components. Polymers 2021, 13(21), 3767.
